# The Effect of a Non-Invasive Positive Pressure Ventilation Simulation Program on General Ward Nurses’ Knowledge and Self-Efficacy

**DOI:** 10.3390/ijerph18062877

**Published:** 2021-03-11

**Authors:** Moon-Sook Kim, Mi-Hee Seo, Jin-Young Jung, Jinhyun Kim

**Affiliations:** 1Medical Nursing Department, Seoul National University Hospital, Seoul 03080, Korea; behappynow@snuh.org (M.-S.K.); iris4@snuh.org (M.-H.S.); amber6@snuh.org (J.-Y.J.); 2College of Nursing, Seoul National University, Seoul 03080, Korea

**Keywords:** simulation-based ventilator training, educational strategy, ventilator-related knowledge, ventilator-related self-efficacy, ventilator nursing

## Abstract

The purpose of this study is to develop a simulation-based ventilator training program for general ward nurses and identify its effects. Quantitative data were collected from 29 nurses (intervention group: 15, control group: 14), of which seven were interviewed with focus groups to collect qualitative data. The quantitative results revealed significant differences in ventilator-related knowledge (*p* = 0.029) and self-efficacy (*p* = 0.026) between the intervention and control groups. Moreover, three themes were derived from meaningful statements in the qualitative data: understanding psychophysical discomfort of the patient while applying the ventilator; helping in ventilator care; and establishing a future ventilator training strategy. The findings confirmed that the non-invasive positive pressure ventilation (NPPV) simulation program is an effective method for improving the knowledge of ventilator nursing and self-efficacy and will be helpful in developing educational methods and strategies related to ventilator nursing for general ward nurses.

## 1. Introduction

Ventilators are conventionally applied with invasive positive pressure through artificial airways such as in tracheostomy or endotracheal intubation in intensive care units [1,2]. However, due to the development of medical devices, and improvement in medical equipment quality and patient survival rates, the number of patients applying for home medical ventilators with excellent performance and portability is continuously increasing [2,3,4,5].

As the number of patients requiring ventilator treatment increased, hospitals faced a lack of beds in intensive care units and an increase in medical expenses in South Korea [2,3,4,5]. This meant that under the judgement of medical staff, many patients who need ventilator treatment received it in general wards after changing to non-invasive positive pressure ventilation (NPPV) [2,3,4].

As the use of home ventilators increases in general wards, the proportion and time of nursing work related to ventilators is also increasing [2,3,4]. However, general ward nurses have fewer educational opportunities and experiences in the ability to use ventilators than nurses in intensive care units, thus making skillful handling of ventilators difficult for the former [1,2,3,4].

For the successful administration of ventilator treatment, not only are a patient’s underlying disease and physical condition important, but their emotional condition is as well [6,7]. Patients who use ventilators often experience the feeling of loss of communication, sleep, self-regulation, self-determination, and personality, and the presence of anguish, fear, and dehumanization [7,8,9]. Particularly, in the case of patients who use home ventilators in general wards, most of them are conscious; therefore, providing emotional care to patients in these wards is more important than providing the same in intensive care units. However, in actual clinical settings, patients’ care mainly focuses on physical care, often leading to the neglect of emotional care [7,8].

Various studies on the usefulness of ventilator treatment and non-invasive positive pressure ventilation (NPPV) have shown that the frequency of intubation in the trachea is low, and there is a reduction in the length of hospitalization, readmission rate, and mortality rate [9,10,11]. However, for the application and maintenance of NPPV, the time and effort required by skilled medical staff are significant factors [7,12,13].

In the UK and Australia, to maintain the continuity of nursing in intensive care units and general wards, respiratory specialists educate and provide knowledge on various types of treatment and skills necessary for medical staff in general wards [14,15]. In South Korea, mainly in large hospitals, there are nurses in charge who assist in the application and management of home ventilators in general wards and also provide education for the medical staff regarding ventilators. However, one existing study analyzed the causes of injuries related to home ventilators among patients in general wards of tertiary hospitals and suggested that systematic and appropriate home ventilator management education for medical staff is necessary [2]. Two other existing studies found that general ward nurses had low nursing knowledge and emergency coping skills for patients with home ventilators, and that there was a high demand for education [3,4]. One of these studies suggests developing a systematic ventilator education program, while the other suggests practical education and monitoring systems based on clinical cases, or alternatively, using simulations [4]. Another study found that more than 70% of medical staff taking care of home ventilators in general wards did not receive systematic home ventilator-related education. These findings clearly illustrate the necessity of providing education to general ward medical staff and suggest that education—including problem-solving and effective analysis for the education program—should be conducted [14].

In this study, a simulation-based ventilator education program including applying NPPV was developed and implemented for nurses in the general ward of internal medicine. The purpose of this study is to analyze the effects of this educational program on knowledge change and self-efficacy in general ward nurses, and their overall educational experience. Through this, we could provide the basic data necessary for the development of a ventilator education program for nurses in general wards and prepare specific ventilator nursing intervention plans.

## 2. Materials and Methods

### 2.1. Research Design

This study is a mixed design study that aims to analyze the effects of NPPV experience and simulation education on the ventilator-related knowledge and self-efficacy of general ward nurses. Quantitative data were collected through standardized questionnaires. Written evaluation papers were prepared and focus group interviews were conducted and analyzed.

### 2.2. Participants and Data Collection

This study involved nurses who were working in the internal medicine ward of a tertiary hospital in Seoul. The notice was put up for two weeks in August 2018, and the research subjects were nurses who were invited to voluntarily participate in the study. The volunteers were assigned to experimental and control groups based on their ventilator nursing experience within the past year. To confirm the bias between the experimental and control groups, homogeneity was verified through a preliminary investigation. The purpose of the study was explained to the participants and informed consent was obtained.

The sample size for the quantitative study was calculated at a significance level of *p* < 0.05, with an effect size of f = 0.5 and power of 1−β = 0.80 by Cohen’s d, resulting in 27 participants in each group, with a dropout rate of 10%. A total of 60 nurses were recruited. However, 31 nurses withdrew participation because of an inability to participate due to workload. Of the 29 nurses who were finally recruited, 15 were assigned to the experimental group and 14 to the control group.

Seven nurses also took part in the qualitative study and they provided their consent to be a part of the focus group interview. The purpose and method of the study were explained to them. The researchers then selected participants and conducted the interviews. In addition, 20 sheets of written evaluation papers were prepared after the above simulation program ended; these were included in the qualitative research analysis data.

### 2.3. Ethics

For the protection of participants in the research, this study received an approval (1709-034-883) from the Ethics Committee of Seoul National University Hospital. After obtaining consent from the participants, the questionnaires and focus group interviews were conducted. It was explained to the participants that they could withdraw their consent any time. The collected data were stored in a designated place so that only the relevant researcher could read it through a locking device, and data input and analysis were coded so that personal identification was impossible. The collected information is expected to be discarded in three years, after the end of the study.

### 2.4. Research Tools

The quantitative research tools of this study were prepared by referring to the safe initiation and management of mechanical ventilation in clinical practice guidelines [16] of the American Association of Respiratory Therapy (AARC) and the ventilator nursing interventions for adult nursing [17]. General ventilator nursing and NPPV management were extracted, except for invasive ventilator management. A respiratory education nurse with more than seven years of experience consulted with one respiratory physician and prepared a preliminary questionnaire. Thereafter, the questionnaire was modified and supplemented with consultations from three nurses with master’s degrees and advanced practice nurses with over five years of clinical experience in a general hospital. Finally, the questionnaire was verified through a group of experts, including two head nurses with several years of experience in ventilator nursing, one professor from the Department of Adult Nursing, and a professor from the department of respiratory medicine.

#### 2.4.1. Nursing Knowledge on the Ventilator

The ventilator nursing knowledge consisted of 10 preliminary questions regarding knowledge necessary for practical performance, such as preparation, maintenance, and management of ventilators. However, items requiring one semantic separation were split into two items during the content validation. Finally, the tool consisted of 11 questions were evaluated on a 10-point scale, with a higher score indicating a higher level of knowledge. The internal validity of the tool was 1.0, and the reliability was Cronbach α = 0.97 in a preliminary survey of 10 nurses in the general ward. The reliability level for this study was Cronbach α = 0.95.

#### 2.4.2. Self-Efficacy on the Ventilator

The self-efficacy in ventilator nursing was evaluated through 15 questions, including topics on confidence, self-regulation, and task difficulty preference at each stage of evaluation, diagnosis and planning, performance, and evaluation of the ventilator nursing process. The final 15 questions were selected through content validation. This tool comprised a total of 15 questions evaluated on a 10-point scale, with a higher score indicating a higher level of self-efficacy. The internal validity of this tool was 1.0, and the reliability was Cronbach’s α = 0.99 in a preliminary survey of 10 subjects. The reliability level for this study was Cronbach’s α = 0.97.

#### 2.4.3. NPPV Experience and Development Process of the Simulation Program

Previous studies on the educational needs of home ventilator nursing [3,14] for general ward nurses and on errors and emergencies associated to ventilator nursing [2,3,4] were used as a basis to draft the program. Frequent ventilator nursing practices performed in general wards and recurring errors were included among the 32 educational needs that were drawn out from the survey contents based on the experiences of a respiratory education nurse. Appropriate items for the general ward situations were chosen by a respiratory nurse and three nurses who were in charge of education. Lastly, six details which included the connecting and management of circuits, explaining the need for a ventilator and use of masks, ventilator manipulation, and treatment of clinical indicators, were reviewed and picked out by three head nurses in a general ward and a professor from the department of respiratory medicine. A one-minute NPPV experience was included in the program to enhance empathy by referring to the papers on the painful experiences of patients with ventilators [18,19].

#### 2.4.4. Contents and Application of the Simulation Program

The operation of the simulation program was conducted for 30 min each time, six times from September 21 to 28. Five to six subjects participated in the program at one time, and two persons joined the program and performed both the patient’s role and the nurse’s role in two or more of the six scenarios. The remaining four scenarios were used to observe the performance of the other groups, and after each scenario was completed, a one-minute debrief was conducted to conduct discussions on each process. Subjects who acted as patients were given a Q card to induce key situations and play them out.

The set value of experiences in the patient role was set to 8 cm H_2_O for inspiration and 4 cm H_2_O for exhalation, the lowest pressure commonly applied in clinical practice, and the actual experience time of applying a face mask was approximately one minute.

### 2.5. Data Collection

This study was conducted through the process shown in Figure 1. All subjects were examined in advance regarding their knowledge of and self-efficacy in ventilator nursing before the training. Both the experimental and control groups received a 30-min lecture on ventilator theory. In the experimental group, post-evaluation was conducted after the theoretical lecture and the simulation program; for the control group, a post-evaluation was conducted after the theoretical lecture, followed by the participation in the simulation program. A 10-min break was given between theory training and the simulation program, during which the control group took a post-evaluation while the experimental group took a break in a separate space followed by the participation in the simulation program. One month after the entire education was completed, a focus group interview was conducted to collect qualitative data.

### 2.6. Data Analysis

#### 2.6.1. Quantitative Analysis

The collected data were analyzed by descriptive statistics. The detailed statistical methods are as follows:
The general characteristics of the experimental group and control group, along with the characteristics related to ventilator care were calculated as real numbers and percentages, and the homogeneity test was analyzed using the *χ*^2^ test;The subjects’ nursing knowledge on ventilators and the degree of self-efficacy were calculated as the mean and standard deviation using the developed tool, and a *t*-test was used for the prior homogeneity test;The difference between the nursing knowledge on ventilators and the degree of self-efficacy between the experimental group and the control group was analyzed using a *t*-test.

#### 2.6.2. Qualitative Analysis

One month after the end of the program, nurses who were willing to participate in the interview and agreed to being recorded were grouped and underwent a 60-min focus group interview. The focus group interview was conducted by the researcher in a quiet conference room and was recorded with a digital recorder. The interview was conducted by the four researchers who had completed a qualitative analysis class in nursing master’s course. The interview process consisted of a beginning, introduction, transition, core, and ending. It proceeded from general questions to specific questions and from positive to negative questions. The structured interview questions are as follows. (1) Introduction question: “How did you feel when experiencing the NPPV?”; (2) key question: “What did you see and feel during the NPPV simulation training?”; (3) sub-questions: “Please tell me if the NPPV simulation training was helpful, and if so, how?”, “What challenges did you face during the NPPV simulation training? Please specify.”, “Have you ever experienced ventilator patient care before and after the NPPV simulation training? If so, please tell us what you experienced”; (4) closing question: “Please tell me if there is anything else you would like to say or suggest for improving ventilator care.”

NPPV experiences and experiences related to ventilator nursing were subject to inductive content analysis [20]. After transcribing the interview content, they were read repeatedly and open-coded in search of meaningful content. After coding and re-reading the data, grouping and categorizing similar concepts, and abstracting semantic units for each subject in consideration of their commonality and relevance, derivation of the core subject were performed. Data analysis was conducted simultaneously with data collection. The four researchers discussed whether the analyzed concept or category was consistent with the participants’ statements. Following this, the results of the qualitative analysis were verified by a professor who majored in qualitative research.

## 3. Results

### 3.1. Participant’s General Characteristics 

Prior to the simulation training, differences between the experimental group and the control group were analyzed in terms of age, sex, education level, work experience, previous ventilator training completion, and the number of ventilator care cases handled within one year.

The clinical work experiences of the subjects varied from less than one year to more than ten years, with most participants having less than three years of clinical work experiences (72.4%). In terms of whether they had previous experiences in ventilator training, a majority (82.8%) of the participants had received ventilator training in the past. Regarding the number of ventilator nursing experiences within one year, most of the participants had none or less than nine years of experience (Table 1).

### 3.2. Participant’s Analysis of Homogeneity

The analysis of the homogeneity between the experimental group and the control group on the degree of self-efficacy and nursing knowledge on ventilator before the NPPV simulation training revealed no significant difference (*p* > 0.05), thus verifying prior homogeneity between the two groups (Table 2).

### 3.3. Changes in the Participant’s Nursing Knowledge and Self-Efficacy

After the ventilation theory education and simulation program, the total score of the ventilation nursing knowledge was 91.53 ± 10.45 in the experimental group, and 79.14 ± 17.80 in the control group; it was significantly higher in the experimental group (*p* = 0.029). There was a significant difference between the experimental and control groups for each item: “I know what to explain to the patient before applying the ventilator” (*p* = 0.009); “I know how to connect the ventilator circuit” (*p* = 0.020); “I know how to manipulate and change the ventilator” (*p* = 0.039); “I know the discomfort and complications caused by the application of the ventilator.” (*p* = 0.011); and “I know what to record related to the ventilator nursing” (*p* = 0.043).

After the simulation program, the total score of the ventilator nursing self-efficacy was 123.73 ± 10.21 in the experimental group, which was significantly higher (*p* = 0.026) than the control group score of 110.00 ± 20.05. Significant differences were observed between the experimental and control groups for each questions: “Ability to assess the indications and necessity of artificial ventilator” (*p* = 0.012); “Ability to determine the priority of problems related to ventilator nursing based on the patient’s health status” (*p* = 0.014); “A nursing diagnosis suitable for nursing with a ventilator can be established on the basis of the nurses’ definition” (*p* = 0.030); “Can explain well the necessity and precautions of applying a ventilator to a patient or family” (*p* = 0.020); “Ability to respond appropriately to artificial ventilator alarms and emergency situations” (*p* = 0.044); “A non-invasive mask can be applied by minimizing discomfort to the patient” (*p* = 0.027); “If necessary, artificial ventilators can be applied which will ensure proficient respiratory manipulation” (*p* = 0.021); and “Respiratory complications and side effects can be prevented” (*p* = 0.017) (Table 3).

### 3.4. Qualitative Data Analysis

After analyzing the data for the development and effects of the simulation program, a total of 51 major statements were extracted. Those with repeated or similar meanings were categorized into 20 subcategories. Finally, three key phrases related to the categories were derived: “to understand the psychological and physical discomfort of the patient applying NPPV”, “there is practical help available in ventilator care”, and “development of ventilator training education and establishment of a strategy”.

Regarding the first theme—understanding the psychological and physical discomfort experienced by patients with ventilator—the nurses expressed that they experienced and understood the psychological and physical discomfort through the simulation program. On the sub-themes of experiencing psychological rejection, subjects expressed feelings of embarrassment, fear, anxiety, and frustration: “I felt frustrated and trapped, and I couldn’t control it, so I think I got a temper later”. Regarding the nurses’ experience of physical discomfort, they complained of physical discomfort such as pain in the face and feeling nausea: “My stomach was full and I felt like throwing up” and “The tightness of the face was more painful and made it difficult to breathe”. Furthermore, the sub-themes of understanding patients with ventilators were expressed as follows: “Sometimes I was annoyed because I couldn’t understand that NPPV patients take off their masks because they feel frustrated and uncomfortable, but when I applied it to myself, I understood 99% of their mind”.

The second theme was that the simulation was helpful in the intervention for ventilator nursing in clinical practice. After participating in the simulation program, they expressed that their fear was reduced, and confidence was gained when caring for patients with ventilators in clinical practice: “After the experience, I saw a patient applying a ventilator, but I felt more confident than before”, and “I don’t think I will be embarrassed even if a patient with NPPV comes”. The other sub-theme was realizing the nursing needs of patients while applying NPPV. Nurses expressed that they realized the need for an explanation and emotional support when applying NPPV through the simulation program: “Every time I tell a patient to be patient, I will understand their mind a little. I thought I had to explain it well in the future”.

The last theme was to establish a future ventilator training strategy. Nurses freely expressed the merits and improvement directions of this simulation program. Regarding the advantages of the simulation program, the nurses expressed that it was practically helpful to perform the necessary manipulations directly through the machine used in the ward and receive feedback in situations that may occur in actual clinical settings; they further stated that it was interesting to have an opportunity to understand patients through their experience. “It was easy to understand by using the machine used in the actual ward. I was familiar with it a lot, but it was nice to learn about the part I did not exactly know”. The second sub-theme was about the notable challenges faced in the program. The nurses expressed regret that the simulation program involved a limited situation in a short time. They also expressed that if there were additional practical instructors, it will be more educational and efficient: “Various patient cases… How did they apply and what happened in a certain situation? What happened to the ABGA (arterial blood gas analysis) value and how did it change? If there was an education about…” (Table 4).

In summary, the simulation program experience was based on three themes: understanding the psychological and physical discomfort of patients while applying NPPV, practical help in the intervention of mechanical ventilator nursing, and establishing a strategy for developing education for mechanical ventilator nursing.

## 4. Discussion

Quantitative results demonstrated that simulation education positively affected the participants’ learning outcomes and work ability in clinical settings [21,22]. It was found that in the nurse group with the simulation program, the degree of knowledge (*p* = 0.029) and self-efficacy (0.026) of ventilator nursing improved significantly. In particular, efficacy in providing explanations necessary for patients after the participation in the simulation program, connection to the ventilator circuit, ventilator operation, prevention of discomfort and side effects, knowledge of nursing records and evaluation of ventilator indications, prioritization, establishment of nursing diagnosis, solving problem situations, mask application, and manipulation showed significant improvement. Among these, patient explanation, manipulation, and side effects prevention items were the focus in six simulation situations, and both nursing knowledge and self-efficacy showed improvement, confirming the effectiveness of the simulation program.

In the preliminary survey, the items with high knowledge score of ventilator nursing were applying a mask to the patient (6.52 ± 1.379) and nursing record (6.38 ± 1.321); the items with a low knowledge score were analysis of ventilator monitoring values (5.48 ± 1.379), followed by the ventilator principle (5.72 ± 1.386). The items with high self-efficacy scores of ventilator nursing were ventilator nursing records (6.62 ± 1.293) and nursing diagnosis establishment (6.48 ± 1.153); the items with low self-efficacy scores were side effect prevention (5.62 ± 1.208) and mask application (5.90 ± 1.448). This is similar to the result of the lowest knowledge score on the prevention of side effects, and setting the mode in the ventilator for nursing students [23]. From this point of view, it was confirmed that the degree of knowledge regarding “applying a mask to a patient” was high but the degree of self-efficacy was low, and education on the application of masks should include more practical education than theory.

The qualitative analysis showed that nurses developed a strategy for developing nursing education for mechanical ventilators, including: “understanding the psychological and physical discomfort of the patients while applying NPPV”, “helping practically in the intervention of mechanical ventilators”, and “establishing a future ventilator training strategy”.

Through the one-minute low-pressure experience in this simulation program, nurses expressed psychological rejection (such as embarrassment, fear, anxiety, and frustration), and physical discomfort (such as mask discomfort, facial pain, and nausea-like symptoms). They expressed that they came to understand the patients’ situation, which they could not understand previously. Unlike nurses who voluntarily participated in the experience, the patient felt resentment and sadness during involuntarily NPPV application [8], skin damage at the contact area of the mask, and positive pressure ventilation caused by continuous application for a long time. It is not possible to understand the patient’s pain because it cannot be experienced as stress-related gastritis. However, understanding patients who were undergoing ventilator treatment means that the sympathy for the patient is a conflict between the patient and the caregiver [24]. It this context, the higher the sympathy of the nurse, the higher the job satisfaction they get.

In addition, the nurses who participated in this simulation program expressed that they became more confident in ventilator care and realized the need for nursing intervention, which supports the results of quantitative studies showing improved knowledge and self-efficacy of ventilator nursing. Previous studies have shown that nurses who effectively coped with stress showed low job stress and increased job satisfaction [25]. This simulation program indicated positive signs for the stress and job satisfaction of nurses caring for patients with ventilators.

The strengths and improvements of this program were derived through the final theme of establishing a strategy for developing an education program on mechanical ventilator nursing. The part presented as a strength is interesting because it allows a real patient experience. It was possible to check changes in situations that could occur while manipulating them, which was directly helpful in practice. This method can be used to increase motivation in future ventilator education. This is supported by previous studies showing that higher learning participation motivation of adult learners leads to higher educational satisfaction and learning outcomes [26]. In addition, if education does not affect practical application, it has no value, and in light of the importance of transferring education into practice [27], the design of this simulation program and the aspect of inducing learners’ motivation are meaningful. However, the lack of time, instructors, and situational aspects mentioned by participating nurses are limitations of this simulation program, and environmental improvement is expected in future ventilator training for general ward nurses.

## 5. Conclusions

The study confirmed that the non-invasive ventilator simulation program, which allows people to directly experience problem-solving situations and patient experiences in a clinical-like environment, is an effective educational method for improving the knowledge of ventilator nursing and self-efficacy.

The study also noted that the simulation program experience was based on three themes: understanding the psychological and physical discomfort of patients while applying NPPV, practical help in the intervention of mechanical ventilator nursing, and establishing a strategy for developing education for mechanical ventilator nursing. Through this, it provides meaningful data on educational methods and strategies to help patients with NPPV, which is increasing in the general ward, and to provide skilled nursing care.

The limitations of this study are as follows. First, in terms of the composition of the educational program, the limitations identified through the written evaluation were a short training time, limited simulation situation, and insufficient instructors. In terms of the operation of the program, it was a one-time short training and not a regular training program, and that the training hours were limited to the hours immediately after work. In terms of research, short-term knowledge and self-efficacy were measured without examining nurses’ practical performance to confirm the effectiveness of the program, and the dropout rate of the participants was high due to the limitation of nurses’ working hours. Through this a follow-up research is suggested. Firstly, it was confirmed that the NPPV simulation program was an effective intervention in increasing nursing knowledge and self-efficacy; however, self-efficacy is closely related to self-directed learning, therefore further research is needed to clarify the relationship. In addition, a future study on repetitive education and its long-term effects is needed because the training effect will be diminished after a period. Secondly, based on the results of this study, we proposed the development of various simulation scenarios and effectiveness evaluation studies related to ventilator nursing. In particular, given the nature of adult education, we proposed to implement several high-quality educational programs and conducting effectiveness assessment studies utilizing experienced instructors and with sufficient training time and simulation training. Thirdly, the measurement tools in this study were created by extracting items suitable for NPPV, which are primarily applied by nurses in general wards; hence, there is a limit to applying these tools to overall ventilator care, including the care of those subjected to invasive ventilation. Therefore, in the future, we proposed to conduct studies on simulation-based learning including various case-based emergency and care for invasive ventilators.

## Figures and Tables

**Figure 1 ijerph-18-02877-f001:**
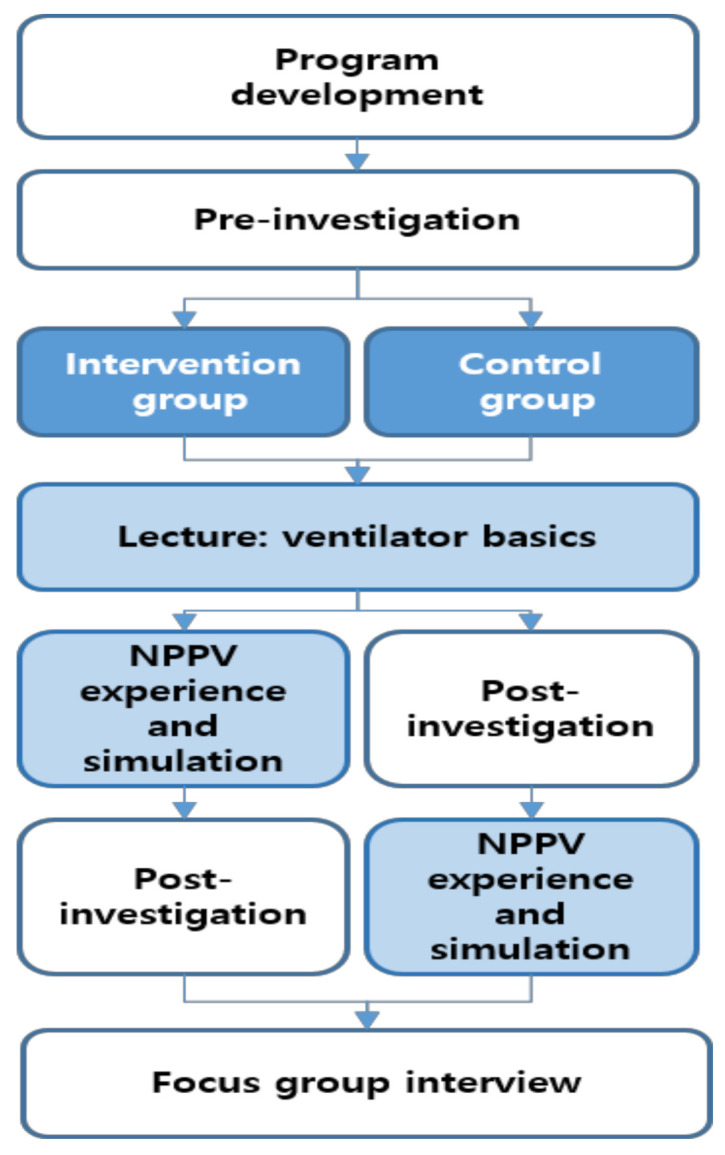
Study procedure. NPPV: non-invasive positive pressure ventilation.

**Table 1 ijerph-18-02877-t001:** Participants’ general characteristics.

Characteristics	Category	Experimental (*n* = 15)	Control (*n* = 14)	*χ* ^2^	*p*
		*n* (%)	*n* (%)		
Age (years)	20–29 30–39 40–49 50–59	13 (86.7) 1 (6.7) 0 (0.0) 1 (6.7)	10 (71. 4) 3 (21.4) 1 (7.1) 0 (0.0)	3.361	0.339
Gender	Female Male	15 (100.0) 0 (0.0)	13 (92.9) 1 (7.1)	1.110	0.292
Education	Associate Bachelor Master	1 (6.7) 12 (80.0) 2 (13.3)	0 (0.0) 14 (100.0) 0 (0.0)	3.123	0.210
Work experience (months)	<12 12–36 37–120 >120	6 (40.0) 4 (26.7) 4 (26.7) 1 (6.7)	3 (21.4) 8 (57.1) 2 (14.3) 1 (7.1)	2.969	0.396
Previous ventilator training	yes no	12 (80.0) 3 (20.0)	12 (85.7) 2 (14.3)	0.166	0.684
Numbers of ventilator nursing within 1 year	0–4 5–9 10–19 >20	5 (33.3) 5 (33.3) 2 (13.3) 3 (20.0)	5 (35.7) 5 (35.7) 2 (14.3) 2 (14.3)	0.166	0.983

**Table 2 ijerph-18-02877-t002:** Participants’ analysis of homogeneity.

Variables	Total (*n* = 29)	Experimental (*n* = 15)	Control (*n* = 14)	*t*	*p*
	Mean ± SD	Mean ± SD	Mean ± SD		
**Degree of ventilator nursing knowledge**	66.76 ± 13.54	69.27 ± 11.86	64.07 ± 15.10	1.034	0.310
1. Indications	6.10 ± 1.448	6.47 ± 1.30	5.71 ± 1.54	1.424	0.166
2. Ventilator mechanism	5.72 ± 1.386	6.00 ± 1.25	5.43 ± 1.50	1.114	0.275
3. Explanation for patients	6.00 ± 1.535	6.40 ± 1.35	5.57 ± 1.65	1.483	0.150
4. Circuit connection	6.00 ± 2.00	6.40 ± 1.59	5.57 ± 2.34	1.105	0.281
5. Troubleshooting	5.83 ± 1.627	6.13 ± 1.64	5.50 ± 1.60	1.049	0.303
6. Application of mask	6.52 ± 1.379	6.80 ± 1.14	6.21 ± 1.57	1.150	0.260
7. Operation	6.24 ± 1.662	6.47 ± 1.55	6.00 ± 1.79	0.750	0.460
8. Discomfort and complications	6.31 ± 1.137	6.53 ± 0.99	6.07 ± 1.26	1.097	0.281
9. Analyze monitoring values	5.48 ± 1.379	5.67 ± 0.90	5.29 ± 1.77	0.737	0.467
10. Nursing record	6.38 ± 1.321	6.47 ± 1.06	6.29 ± 1.59	0.364	0.719
11. Prevention of infection	6.17 ± 1.605	5.93 ± 1.58	6.43 ± 1.65	−0.826	0.416
**Degree of self-efficacy for ventilator nursing**	92.44 ± 17.26	92.26 ± 15.25	92.64 ± 19.78	−0.058	0.955
1. Indication assessment	6.07 ± 1.132	6.00 ± 1.00	6.14 ± 1.29	−0.334	0.741
2. Prioritization	6.00 ± 1.134	5.87 ± 1.12	6.14 ± 1.16	−0.649	0.522
3. Establishing nursing diagnosis	6.48 ± 1.153	5.53 ± 0.83	6.43 ± 1.45	0.236	0.816
4. Establishing nursing plan	6.45 ± 1.183	6.53 ± 0.91	6.36 ± 1.44	0.395	0.696
5. Checking the setting values	6.10 ± 2.006	6.27 ± 1.83	5.93 ± 2.23	0.447	0.658
6. Explaining to patient	6.14 ± 1.407	6.13 ± 1.12	6.14 ± 1.70	−0.018	0.986
7. Preparing materials	6.17 ± 1.627	6.13 ± 1.68	6.21 ± 1.62	−0.132	0.896
8. Troubleshooting	5.93 ± 1.624	5.87 ± 1.64	6.00 ± 1.66	−0.217	0.830
9. Applying mask	5.90 ± 1.448	5.87 ± 1.50	5.93 ± 1.43	−0.113	0.911
10. Operation	5.93 ± 1.831	6.00 ± 1.60	5.86 ± 2.10	0.260	0.838
11. Prevention of complications	5.62 ± 1.208	5.67 ± 1.23	5.57 ± 1.22	0.209	0.836
12. Nursing record	6.62 ± 1.293	6.67 ± 1.11	6.57 ± 1.50	0.195	0.847
13. Cooperation with other medical staff	6.34 ± 1.344	6.13 ± 1.35	6.57 ± 1.34	−0.874	0.390
14. Evaluation after application	6.34 ± 1.143	6.33 ± 0.97	6.36 ± 1.33	−0.055	0.956
15. Modification of nursing plan	6.34 ± 1.143	6.27 ± 1.10	6.43 ± 1.22	−0.375	0.710

**Table 3 ijerph-18-02877-t003:** Changes in the participants’ knowledge and self-efficacy.

Variables	Experimental (*n* = 15)	Control (*n* = 14)	*t*	*p*
	Mean ± SD	Mean ± SD		
**Degree of ventilator nursing knowledge**	91.53 ± 10.45	79.14 ± 17.80	2.305	0.029
1. Indications	7.66 ± 1.29	6.78 ± 1.57	1.651	0.110
2. Ventilator mechanism	8.00 ± 1.13	6.92 ± 1.73	1.986	0.057
3. Explanation for patients	8.33 ± 1.34	6.78 ± 1.62	2.801	0.009
4. Circuit connection	8.60 ± 0.91	7.14 ± 1.95	2.543	0.020
5. Troubleshooting	8.46 ± 1.12	7.35 ± 1.82	1.987	0.057
6. Application of mask	8.53 ± 1.35	7.42 ± 1.65	1.975	0.059
7. Operation	8.73 ± 0.79	7.50 ± 1.91	2.239	0.039
8. Discomfort and complications	8.40 ± 0.98	7.07 ± 1.59	2.723	0.011
9. Analyze monitoring values	7.93 ± 1.09	7.00 ± 1.56	1.866	0.073
10. Nursing record	8.40 ± 0.73	7.42 ± 1.65	2.071	0.048
11. Prevention of infection	8.46 ± 1.24	7.71 ± 1.68	1.374	0.181
**Degree of self-efficacy for ventilator nursing**	123.73 ± 10.21	110.00 ± 20.05	2.348	0.026
1. Indication assessment	8.20 ± 0.77	7.00 ± 1.51	2.708	0.012
2. Prioritization	8.06 ± 0.88	6.85 ± 1.51	2.653	0.014
3. Establishing nursing diagnosis	8.26 ± 0.88	7.28 ± 1.38	2.293	0.030
4. Establishing nursing plan	7.93 ± 1.03	7.21 ± 1.42	1.565	0.129
5. Checking the setting values	8.26 ± 1.16	7.35 ± 1.39	1.370	0.182
6. Explaining to patient	8.40 ± 0.91	7.35 ± 1.33	2.471	0.020
7. Preparing materials	8.26 ± 1.16	7.35 ± 1.39	1.914	0.066
8. Troubleshooting	8.20 ± 0.86	7.21 ± 1.57	2.108	0.044
9. Applying mask	8.13 ± 0.74	7.21 ± 1.31	2.343	0.027
10. Operation	8.66 ± 0.81	7.50 ± 1.55	2.501	0.021
11. Prevention of complications	8.20 ± 0.77	7.28 ± 1.13	2.544	0.017
12. Nursing record	8.33 ± 0.97	7.57 ± 1.39	1171	0.099
13. Cooperation with other medical staff	8.26 ± 0.79	7.57 ± 1.39	1.629	0.119
14. Evaluation after application	8.26 ± 0.70	7.50 ± 1.34	1.904	0.072
15. Modification of nursing plan	8.26 ± 0.70	7.50 ± 1.34	1.904	0.072

**Table 4 ijerph-18-02877-t004:** Analyzing experience of the noninvasive positive pressure ventilator simulation program.

Theme	Sub-Theme	Meaningful Statements
Understanding psychophysical discomfort of the patient while applying NPPV	Experience psychological rejection	Embarrassment, Fear, Anxiety, Closeness
	Experience physical discomfort	Pressure on the face, Pain Feeling of nausea
	Understanding the patient while applying NPPV	Understanding the patient while applying the ventilator
Help in ventilator care	Build confidence for ventilator care	Reduced anxiety Increased confidence
	Realizing the nursing needs of patient while applying NPPV	Needs for emotional support needs for explanation
Establishing a future ventilator training strategy	Strengths of the program	Interest, Helps in clinical practice, Way to understand patients while applying NPPV, Realism
	Notable challenges of the program	Lack of time, Lack of the number of instructors, Limitation of practice situations

## Data Availability

The data presented in this study are available on request from the corresponding author.

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
