# Peer review of "The Effect of a Non-Invasive Positive Pressure Ventilation Simulation Program on General Ward Nurses’ Knowledge and Self-Efficacy"

_ijerph, 2021, doi:10.3390/ijerph18062877_

Round 1
Reviewer 1 Report
Thank the authors for their hard work in writing this article. There are some parts that need improvement.
Most references can be updated.
Line 41 :’ Since general ward nurses have fewer educational opportunities and experience in the ability to use ventilators than nurses in intensive care units, skillful handling of ventilators is not an easy for the former’. Please add a reference?
Participant: Please provide more details on how, why, when and where to choose participants? Random? Or convenient sampling?
Please add detailed information about the qualitative data collected and focus group procedures. Who, how and how? In lines 170-177, "Four researchers were discussed and analyzed." Please also indicate the details of the background and qualitative research experience of these four researchers.
Two questionnaires (1) Nursing knowledge on the ventilator(2) Self-efficacy on the ventilator were used in the study, which designed by the same group of experts? . and the internal validity of the tools both were 1.0? Please add more details about the internal validity of these three experts? For example: Frontier question 3 ‘Explanation for patients’, since this study aims to improve the general ward nurses’ knowledge about using NPPV in order to provide the better quality of care to the patients. However, the interpretation of patients depends on subjective belief that nurses are really good explain it or ‘giving patient- center objectively good explanation? Therefore, who and how to determine the validity of the expert internship as 1.0? same problem as the set (2) question 6.
Line 119: (3) Are the experience of NPPV and the development process of the simulation program also designed by the same expert group mentioned above? How to choose those expert groups, who and how?
Line 123: "Based on the content of the survey on the educational needs of nurses in the general ward for nursing home 123 ventilator, a draft was drafted." The article published by the different authors in 2012 is the latest education needs of Korean general ward nurses? Since the knowledge and skills required for innovative medical conditions vary from day to day, it is important to ensure that it is up to date.
Line 142-146 data collection: experimental and control group’ participants were working in the same hospital and whether both conducted at the same time? Same condition and who deliver the stimulation program? Same educators? how to prevent the data contamination?
Lines 236-247: The title is "effectiveness"...refers to the degree to which something successfully produces the desired result; it should test measurable items or reliable indicators, but the author used "self-efficacy" , it means that we have confidence in our own abilities, especially our ability to meet the challenges before us and successfully complete tasks. Please explain in detail the relationship between the above two concepts.
From the discussion section: applying a mask to the patient (6.52±1.379) and nursing record (6.38±1.321); the items with a low knowledge score were: analysis of ventilator monitoring values
(5.48±1.379), followed by the ventilator principle (5.72±1.386). The items with high self-efficacy scores of ventilator nursing were: ventilator nursing records (6.62±1.293) and nursing diagnosis establishment (6.48±1.153); the items with low self-efficacy scores were side effects prevention (5.62±1.208) and mask application (5.90±1.448) From the score presents the nurses in the skills of routine work for example: wearing mask or record could be kept well but on the based knowledge about ventilator monitoring values ventilator monitoring values is rather poor, the self- efficacy on the score in mask application also was poor, two set of questionnaires but different truth, needs more explanation.
Author Response
Reply to Reviewers:
Thank you for the comments and helpful suggestions regarding our manuscript. We have modified the manuscript accordingly and the details of the revisions are provided below.
Reviewer reports:
Reviewer 1
- Line 41: Since general ward nurses have fewer educational opportunities and experience in the ability to use ventilators than nurses in intensive care units, skillful handling of ventilators is not an easy for the former’. Please add a reference?
Response: I have added a reference to take your comment into account.
- (Before) Since general ward nurses have fewer educational opportunities and experience in the ability to use ventilators than nurses in intensive care units, skillful handling of ventilators is not an easy for the former.
- (After) However, general ward nurses have fewer educational opportunities and experience in the ability to use ventilators than nurses in intensive care units, thus making skillful handling of ventilators difficult for the former [1-4].
- Participant: Please provide more details on how, why, when and where to choose participants? Random? Or convenient sampling?
Response: (L82-88) I have revised the details on the participants according to your comment.
- (Before) This study included nurses who worked in the internal medicine ward of the third General Hospital in Seoul. The purpose of the study was explained to them and consent was taken.
- (After) This study included nurses working in the internal medicine ward of a tertiary hospital in Seoul. The notice was put up for two weeks in August 2018, and the research subjects were nurses who were invited to voluntarily participate in the study. The volunteers were assigned to experimental and control groups based on ventilator nursing experience within the past year. To confirm the bias between the experimental and control groups, homogeneity was verified through a preliminary investigation. The purpose of the study was explained to the participants and informed consent was obtained.
- Please add detailed information about the qualitative data collected and focus group procedures. Who, how and how? In lines 170-177, "Four researchers were discussed and analyzed." Please also indicate the details of the background and qualitative research experience of these four researchers.
Response: (L201-224) I have added detailed information based on your comment.
.
- (Before) NPPV experiences and experiences related to ventilator nursing were subjected to inductive content analysis [15]. After transcribing the written questionnaire and interview content, they were read repeatedly and open-coded in search of meaningful content. After coding and re-reading the data, grouping and categorizing similar concepts, and abstracting semantic units for each subject in consideration of their commonality and relevance, derivation of the core subject was carried out. Data analysis was conducted simultaneously with the collection, and four researchers discussed whether the analyzed concept or category was consistent with the participants' statements.
- (After) One month after the end of the program, nurses who were willing to participate in the interview and agreed to being recorded were grouped and underwent 60-minute focus group interviews. The focus group interview was conducted by the researcher in a quiet conference room and recorded with a digital recorder with the consent of the participant. The four researchers were those who had completed the qualitative analysis class in the master’s course in nursing. The interview process consisted of a beginning, introduction, transition, core, and ending, and proceeded from a general question to a specific question, and from positive to negative questions. The structured interview questions are as follows. (1) Introduction Question: “How did you feel when experiencing the NPPV?” (2) Key question: What did you see and feel during the NPPV simulation training? (3) Sub-question: "Please tell me if the NPPV simulation training was helpful, and if so, how?", “What challenges did you face during the NPPV simulation training? Please specify.”, “Have you ever experienced ventilator patient care before and after the NPPV simulation training? If so, please tell us what you experienced.” (4) Closing Question: “Please tell me if there is anything else you would like to say or suggest for improving ventilator care.”
NPPV experiences and experiences related to ventilator nursing were subjected to inductive content analysis [21]. After transcribing the written questionnaire and interview content, they were read repeatedly and open-coded in search of meaningful content. After coding and re-reading the data, grouping and categorizing similar concepts, and abstracting semantic units for each subject in consideration of their commonality and relevance, derivation of the core subject was performed. Data analysis was conducted simultaneously with data collection, and four researchers discussed whether the analyzed concept or category was consistent with the participants’ statements. Following this, the results of the qualitative analysis were verified by a professor who majored in qualitative research.
- Two questionnaires (1) Nursing knowledge on the ventilator (2) Self-efficacy on the ventilator were used in the study, which designed by the same group of experts? and the internal validity of the tools both were 1.0? Please add more details about the internal validity of these three experts? For example: Frontier question 3 ‘Explanation for patients’, since this study aims to improve the general ward nurses’ knowledge about using NPPV in order to provide the better quality of care to the patients. However, the interpretation of patients depends on subjective belief that nurses are really good explain it or ‘giving patient-center objectively good explanation? Therefore, who and how to determine the validity of the expert internship as 1.0? same problem as the set (2) question 6.
Response: (L112-124) I have added the content and modified this text per your suggestion.
- (Before) The quantitative research tool of this study was prepared by referring to the safe initiation and management of mechanical ventilation [11] among the ventilator nursing interventions of adult nursing [12] and the Clinical Practice Guidelines of the American Association for Respiratory Care (AARC). It was produced through the process of revision 98 with expert advice.
- (After) The quantitative research tools of this study were prepared by referring to safe initiation and management of mechanical ventilation in clinical practice guidelines [17] of the American Association of Respiratory Therapy (AARC) and the ventilator nursing interventions for adult nursing [18]. Among them, general ventilator nursing and NPPV management were extracted, except for invasive ventilator management. A respiratory education nurse with more than seven years of experience consulted with one respiratory physician and prepared a preliminary Thereafter, the questionnaire was modified and supplemented with consultations from three nurses with master’s degrees and advanced practice nurses with over five years of clinical experience in a general hospital. Finally, the questionnaire was verified through a group of experts, including two head nurses with several years of experience in ventilator nursing, one professor from the Department of Adult Nursing, and one professor from the department of respiratory medicine.
Response: (L126-129) I have added the content and modified this text per your suggestion.
- (Before) As for the nursing ventilator related knowledge measurement tool, two head nurses with significant experience in ventilating nursing got the questions filled out by the following individuals: a nurse in charge of breathing education, one professor in the Department of Adult Nursing, and one professor in the department of respiratory medicine to evaluate the validity of the questions and correct parts with ambiguous meaning.
- (After) Ventilator nursing knowledge consisted of 10 preliminary questions regarding knowledge necessary for practical performance, such as preparation, maintenance, and management of ventilators; the items that needed semantic separation were separated and described during content validity verification.
Response: (L135-138) I have added the content and modified this text as you suggested.
- (Before) The self-efficacy measurement tool on a ventilator was corrected by a nurse in charge of breathing education. This was done after filling out the questionnaire by two general ward head nurses with significant experience in ventilating nursing, one adult nursing professor, and one respiratory internal medicine professor.
- (After) The self-efficacy in ventilator nursing was evaluated through 15 preliminary questions, including on confidence, self-regulation, and task difficulty preference at each stage of evaluation, diagnosis and planning, performance, and evaluation of the ventilator nursing process. The final 15 questions were selected through content validation.
- Line 119: (3) Are the experience of NPPV and the development process of the simulation program also designed by the same expert group mentioned above? How to choose those expert groups, who and how?
Response: (L145-158) I have revised the content taking your comment into account.
- (Before) Based on the results of the study [13-14], the experience of patients with NPPV was included in the simulation program as a way to improve communication in the process of understanding the fear and rejection felt by patients, and forming empathy. In addition, a draft was composed based on the contents of the survey on educational needs for home ventilator nursing for general ward nurses [9]. The simulation program development was organized based on the draft of four educational nurses who had experienced simulation education for many years. A total of six core contents were selected based on nursing behaviors necessary in situations often encountered in clinical practice, and nursing behaviors that are prone to errors. These were reviewed by four nurses, three head nurses, and one professor of respiratory medicine.
- (After) Based on previous studies [19-20], the experience of patients with NPPV was included in the simulation program as a way of improving communication in the process of understanding the fear and rejection experienced by patients, and the formation of empathy. The draft of the program was based on prior studies on the educational needs of home ventilator nursing for general ward nurses and on errors and emergencies related to ventilator nursing [2-4]. A total of 32 educational needs were extracted in the survey contents [15]; respirator nursing practices frequently performed in general wards and errors with a tendency to occur frequently were listed according to the experience of the respiratory education nurse. Finally, one respiratory education nurse and three nurses in charge of education selected items suitable for general ward situations, and three head nurses in a general ward and one professor from the department of respiratory medicine reviewed and selected six key details, including connecting and management of circuits, explaining the need for a ventilator and use of masks, ventilator manipulation, and treatment of clinical indicators.
- Line 123: "Based on the content of the survey on the educational needs of nurses in the general ward for nursing home ventilator, a draft was drafted." The article published by the different authors in 2012 is the latest education needs of Korean general ward nurses? Since the knowledge and skills required for innovative medical conditions vary from day to day, it is important to ensure that it is up to date.
Response: I have added the latest study on the training needs of ward nurses on ventilators.
- Line 142-146 data collection: experimental and control group’ participants were working in the same hospital and whether both conducted at the same time? Same condition and who deliver the stimulation program? Same educators? how to prevent the data contamination?
Response: (L172-181) I have added the content you suggested.
- (Before) This study was conducted through the following process (Figure 1). All subjects were examined in advance regarding knowledge and self-efficacy of ventilator nursing before training, followed by a investigation after 30 minutes of ventilator theory training in the control group, and finally after all ventilation theory training and simulation programs in the experimental group were completed. Additionally, a follow-up investigation was conducted. The control group participated in the simulation program investigation, and both experimental groups participated in the simulation program. Moreover, written evaluation papers were prepared immediately after training.
- (After) This study was conducted through the process shown in Figure 1. All subjects were examined in advance regarding their knowledge of and self-efficacy in ventilator nursing before the training. Both the experimental and control groups received a 30-minute lecture on ventilator theory. In the experimental group, post-evaluation was conducted after the entire theoretical lecture and the simulation program; for the control group, a post-evaluation was conducted after the theoretical lecture, followed by participation in the simulation program. A 10-minute break was given between theory training and the simulation program, during which the control group took a post-evaluation and the experimental group took a break in a separate space followed by participation in the simulation program.
- Lines 236-247: The title is "effectiveness"...refers to the degree to which something successfully produces the desired result; it should test measurable items or reliable indicators, but the author used "self-efficacy", it means that we have confidence in our own abilities, especially our ability to meet the challenges before us and successfully complete tasks. Please explain in detail the relationship between the above two concepts.
Response: (L405-426) To determine the actual effect, the degree of performance directly applied to the patient should be checked, but the fact that it is not possible is a limitation of this study. I have added the limitation of this study.
- (Before) The limitations of this study are that the educational program is only a one-time and short-term effect, and the training time, the number of instructors, and the simulation situations are limited. This study proposes the following directions for future studies. Despite the confirmation of the NPPV simulation program being an effective intervention for increasing nursing knowledge and self-efficacy, further studies on long-term effects are needed. Second, based on the results of this study, developing various simulation scenarios related to ventilator nursing and to evaluate their effectiveness is necessary. In particular, in view of the characteristics of adult education, we propose that a high-quality educational program development and effectiveness evaluation study be conducted multiple times rather with sufficient education time and instructors who are familiar with simulation education
- (After) The limitations of this study are as follows. First, in terms of the composition of the educational program, the limitations identified through the written evaluation were short training time, limited simulation situation, and fewer instructors. In terms of the operation of the program, there were limitations in that it was a one-time short training and not a regular training program, and that the training hours were limited to the hours immediately after work. In terms of research, the limitations were that short-term knowledge and self-efficacy were measured without examining nurses’ practical performance to confirm the effectiveness of the program, and the dropout rate of the participants was high due to the limitation of nurses’ working hours. Through this, the following follow-up research is suggested. First, it was confirmed that the NPPV simulation program was an effective intervention in increasing nursing knowledge and self-efficacy; however, a future study on repetitive education and its long-term effects is needed because the training effect is reduced after a period. Second, based on the results of this study, we propose the development of various simulation scenarios and effectiveness evaluation studies related to ventilator nursing. In particular, given the nature of adult education, we propose implementing several high-quality educational programs and conducting effectiveness assessment studies through instructors familiar with sufficient training time and simulation training. Third, the measurement tools in this study were created by extracting items suitable for NPPV, which are primarily applied by nurses in general wards; hence, there is a limit to applying these tools to overall ventilator care, including the care of those subjected to invasive ventilation. Therefore, in future, we propose studies on simulation based learning including various case-based emergency and care for invasive ventilators.
Response: (L2-4) I have revised the title to take your comment into account.
- (Before) Title: Developing and Analyzing the effectiveness of a Non-Invasive Positive Pressure Ventilation (NPPV) Simulation Program for General ward Nurses
- (After) Title: The Effect of a Non-Invasive Positive Pressure Ventilation Simulation Program on General Ward Nurses’ Knowledge and Self-Efficacy
- From the discussion section: applying a mask to the patient (6.52±1.379) and nursing record (6.38±1.321); the items with a low knowledge score were: analysis of ventilator monitoring values (5.48±1.379), followed by the ventilator principle (5.72±1.386). The items with high self-efficacy scores of ventilator nursing were: ventilator nursing records (6.62±1.293) and nursing diagnosis establishment (6.48±1.153); the items with low self-efficacy scores were side effects prevention (5.62±1.208) and mask application (5.90±1.448) From the score presents the nurses in the skills of routine work for example: wearing mask or record could be kept well but on the based knowledge about ventilator monitoring values ventilator monitoring values is rather poor, the self-efficacy on the score in mask application also was poor, two set of questionnaires but different truth, needs more explanation.
Response: (L352-355) I have modified the contents.
- (Before) It could be confirmed that education for Korean students is needed. On the other hand, in terms of the degree of knowledge of respirator-applied nursing, and the degree of self-efficacy that can be applied in practice, it was confirmed that there is a need for practical education rather than theory.
- (After) From this point of view, it was confirmed that the degree of knowledge regarding “applying a mask to a patient” was high but the degree of self-efficacy was low, and education on the application of masks should include more practical education than theory.

Reviewer 2 Report
The paper entitled "Development and effectiveness of non-invasive positive pressure ventilation (NIPPV) simulation program for general ward nurses" presents an attractive study that could be published. The manuscript exposes an appropriate research problem, indicates the contribution of the work and its objective. In general, it presents a good structure, the results are indicated appropriately, and the conclusions are consistent with the paper's objective. However, the paper has some shortcomings, mainly in the explanation of some procedures related to the data collection process and, also, with the training process that is carried out.
Abstract
The abstract has an adequate structure. However, it may be necessary to indicate the critical conclusions of the study.
- Introduction
The introduction is correct in its structure and content. However, literature related to learning through training could be added, especially that related to the context of public health and, in particular, nursing.
- Materials and method
2.1. Research Design
Correct.
2.2. Participants and Data Collection
Was any study done to check for possible bias in the nurses who did not participate in the study?
The last paragraph of this point needs new wording.
2.3. Ethics
Correct.
2.4. Research Tools
The review process outlined in the first paragraph on page 3 should be explained in more detail.
(1) Nursing knowledge on ventilator
Likert scales are usually odd. Therefore, it might be convenient to eliminate the concept of "Likert."
(3) NIPPV experience and development process of the simulation program
At this point, it could be explained in greater detail how the analysis of training needs was carried out and the justification of the training methodology that was followed.
2.5. Data collection
Correct.
2.6. Data analysis
2) Qualitative analysis
This type of methodology used should be explained and justified in more detail.
- Results
The presentation is correct.
- Discussion
The first paragraph on page 9, "Through the ..." needs further support from the literature. Besides, aspects related to the evaluation of training should be studied in-depth, especially, some strategies aimed at improving the training that has been carried out, and possible problems in applying training to the job could be pointed out.
Study limitations should be indicated in a separate section. Besides, literature should be provided that would open new lines of research.
Author Response
Reply to Reviewers:
Thank you for the comments and helpful suggestions regarding our manuscript. We have modified the manuscript accordingly and the revised details are provided below.
Reviewer reports:
Reviewer 2
Comments and Suggestions for Authors
The paper entitled "Development and effectiveness of non-invasive positive pressure ventilation (NIPPV) simulation program for general ward nurses" presents an attractive study that could be published. The manuscript exposes an appropriate research problem, indicates the contribution of the work and its objective. In general, it presents a good structure, the results are indicated appropriately, and the conclusions are consistent with the paper's objective. However, the paper has some shortcomings, mainly in the explanation of some procedures related to the data collection process and, also, with the training process that is carried out.
Abstract
The abstract has an adequate structure. However, it may be necessary to indicate the critical conclusions of the study.
Response: (L19-22) I have revised the content you suggested.
- (Before) This study will help develop educational methods and strategies for ventilator nursing for general ward nurses.
- (After) The findings confirmed that NPPV simulation program is an effective method for improving the knowledge of ventilator nursing and self-efficacy and will be helpful in developing educational methods and strategies related to ventilator nursing for general ward nurses.
- Introduction
The introduction is correct in its structure and content. However, literature related to learning through training could be added, especially that related to the context of public health and, in particular, nursing.
Response: (L27-64) I have added the content and modified this text per your suggestion.
- (Before) Ventilators are traditionally applied with invasive positive pressure ventilation through artificial airways such as tracheostomy or endotracheal intubation. However, in the initial ventilator treatment, non-invasive positive pressure ventilation (NPPV) is increasingly applied as a method to reduce the risk of having to secure the airway through artificial airway intubation, as well as complications such as vocal cord injury, and infection due to invasive causes [1].
For successful ventilator treatment, the patient's underlying disease, physical condition, and emotional condition are important [2]. Patients who use ventilators often experiences feeling of loss of communication, loss of sleep, loss of self-regulation, loss of self-determination, loss of personality, the presence of pain and fear, and dehumanization [3-4]. In particular, in the case of patients who use home ventilators in general wards, most of them are conscious; therefore, providing emotional care in these wards is much more important than in intensive care units. However, in actual clinical settings, the patients’ care mainly focuses on physical care, often leading to the neglect of emotional nursing care [3].
Since general ward nurses have fewer educational opportunities and experience in the ability to use ventilators than nurses in intensive care units, skillful handling of ventilators is not an easy for the former. Various studies on the usefulness of NPPV have shown that the frequency of intubation in the trachea is low and there is a reduction in the length of hospitalization, readmission rate, and mortality rate [1,5-6]. However, for the application and maintenance of NPPV, the time and effort required by skilled medical staff are important [7-8].
In the UK and Australia, in order to maintain the continuity of nursing in intensive care units and general wards, respiratory specialist educate and support various types of treatment knowledge and skills for medical staff in general wards [9-10]. There have been studies on topics such as the experience of patients who applied the ventilator, the educational needs of patients, the actual status of ventilator training, the knowledge and education requirements, and the experience of patients with the ventilator for the intensive care staff. Few studies have evaluated the effects of NPPV application experience and simulation education for general ward nurses.
- (After) Ventilators are conventionally applied with invasive positive pressure through artificial airways such as in tracheostomy or endotracheal intubation in intensive care units [1-2]. However, due to the development of medical devices, and improvement in medical equipment quality and patient survival rates, the number of patients applying for home medical ventilators with excellent performance and portability is continuously increasing [2-5].
- As the number of patients requiring ventilator treatment increased, hospitals faced a lack of beds in intensive care units and an increase in medical expenses in South Korea [2-5]. It means that under the judgement of medical staff, many patients who need ventilator treatment are receiving that in general wards after changing to NPPV [2-4].
- As the use of home ventilators increases in general wards, the proportion and time of nursing work related to ventilators is also increasing [2-4]. However, general ward nurses have fewer educational opportunities and experience in the ability to use ventilators than nurses in intensive care units, thus making skillful handling of ventilators difficult for the former [1-4].
- For the successful administration of ventilator treatment, not only are a patient's underlying disease and physical condition important but also their emotional condition [6-7]. Patients who use ventilators often experience feeling of a loss of communication, sleep, self-regulation, self-determination, and personality, and the presence of anguish, fear, and dehumanization [7-9]. In particular, in the case of patients who use home ventilators in general wards, most of them are conscious; therefore, providing emotional care to patients in these wards is more important than providing that in intensive care units. However, in actual clinical settings, patients’ care mainly focuses on physical care, often leading to the neglect of emotional care [7-8].
- Various studies on the usefulness of ventilator treatment and non-invasive positive pressure ventilation (NPPV) have shown that the frequency of intubation in the trachea is low, and there is a reduction in the length of hospitalization, readmission rate, and mortality rate [9-11]. However, for the application and maintenance of NPPV, the time and effort required by skilled medical staff are significant factors [7, 13-14].
- In the UK and Australia, to maintain the continuity of nursing in intensive care units and general wards, respiratory specialists educate and provide knowledge on various types of treatment and skills necessary for medical staff in general wards [15-16]. In South Korea, mainly in large hospitals, there are nurses in charge who assist in the application and management of home ventilators in general wards, and provide education for the medical staff regarding ventilators. However, there are few studies on the methods and effects of such an educational program. Most studies conducted in Korea focus on the problems and educational needs of ward nurses resulting from a lack of ventilator education, and an education program for ward nurses is necessary [2-4].
- Materials and method
2.2. Participants and Data Collection
Was any study done to check for possible bias in the nurses who did not participate in the study? The last paragraph of this point needs new wording.
Response: (L82-88) I have revised the details on participants taking your comment into consideration.
- (Before) This study included nurses who worked in the internal medicine ward of the third General Hospital in Seoul. The purpose of the study was explained to them and consent was taken.
- (After) This study involved nurses working in the internal medicine ward of a tertiary hospital in Seoul. The notice was put up for two weeks in August 2018, and the research subjects were nurses who were invited to voluntarily participate in the study. The volunteers were assigned to experimental and control groups based on ventilator nursing experience within the past year. To confirm the bias between the experimental and control groups, homogeneity was verified through a preliminary investigation. The purpose of the study was explained to the participants and informed consent was obtained.
2.4. Research Tools
The review process outlined in the first paragraph on page 3 should be explained in more detail.
(1) Nursing knowledge on ventilator
Likert scales are usually odd. Therefore, it might be convenient to eliminate the concept of "Likert."
Response: (L129-130, 138-139) I have revised the content based on your comment.
- (Before) This tool consists of a total of 11 questions and is evaluated on a 1-10 Likert scale,
- (After) This tool consisted of 11 questions and evaluated on a 10-point scale,
- (Before) This tool consists of a total of 15 questions and is evaluated on a 1-10 Likert scale,
- (After) This tool comprised a total of 15 questions evaluated on a 10-point scale,
(3) NIPPV experience and development process of the simulation program. At this point, it could be explained in greater detail how the analysis of training needs was carried out and the justification of the training methodology that was followed.
Response: (Lines 145-158) I have revised the content according to your comment.
- (Before) Based on the results of the study [13-14], the experience of patients with NPPV was included in the simulation program as a way to improve communication in the process of understanding the fear and rejection felt by patients, and forming empathy. In addition, a draft was composed based on the contents of the survey on educational needs for home ventilator nursing for general ward nurses [9]. The simulation program development was organized based on the draft of four educational nurses who had experienced simulation education for many years. A total of six core contents were selected based on nursing behaviors necessary in situations often encountered in clinical practice, and nursing behaviors that are prone to errors. These were reviewed by four nurses, three head nurses, and one professor of respiratory medicine.
- (After) Based on previous studies [19-20], the experience of patients with NPPV was included in the simulation program as a way of improving communication in the process of understanding the fear and rejection experienced by patients, and the formation of empathy. The draft of the program was based on prior studies on the educational needs of home ventilator nursing for general ward nurses and on errors and emergencies related to ventilator nursing [2-4]. A total of 32 educational needs were extracted in the survey contents [15]; respirator nursing practices frequently performed in general wards and errors with a tendency to occur frequently were listed according to the experience of the respiratory education nurse. Finally, one respiratory education nurse and three nurses in charge of education selected items suitable for general ward situations, and three head nurses in a general ward and one professor from the department of respiratory medicine reviewed and selected six key details, including connecting and management of circuits, explaining the need for a ventilator and use of masks, ventilator manipulation, and treatment of clinical indicators.
2.6. Data analysis
2) Qualitative analysis
This type of methodology used should be explained and justified in more detail.
Response: (L201-224) I have added detailed information according to your comment.
.
- (Before) NPPV experiences and experiences related to ventilator nursing were subjected to inductive content analysis [15]. After transcribing the written questionnaire and interview content, they were read repeatedly and open-coded in search of meaningful content. After coding and re-reading the data, grouping and categorizing similar concepts, and abstracting semantic units for each subject in consideration of their commonality and relevance, derivation of the core subject was carried out. Data analysis was conducted simultaneously with the collection, and four researchers discussed whether the analyzed concept or category was consistent with the participants' statements.
- (After) One month after the end of the program, nurses who were willing to participate in the interview and agreed to being recorded were grouped and underwent 60-minute focus group interviews. The focus group interview was conducted by the researcher in a quiet conference room and recorded with a digital recorder with the consent of the participant. The four researchers were those who had completed the qualitative analysis class in the master’s course in nursing. The interview process consisted of a beginning, introduction, transition, core, and ending, and proceeded from a general question to a specific question, and from positive to negative questions. The structured interview questions are as follows. (1) Introduction Question: “How did you feel when experiencing the NPPV?” (2) Key question: What did you see and feel during the NPPV simulation training? (3) Sub-question: "Please tell me if the NPPV simulation training was helpful, and if so, how?", “What challenges did you face during the NPPV simulation training? Please specify.”, “Have you ever experienced ventilator patient care before and after the NPPV simulation training? If so, please tell us what you experienced.” (4) Closing Question: “Please tell me if there is anything else you would like to say or suggest for improving ventilator care.”
NPPV experiences and experiences related to ventilator nursing were subjected to inductive content analysis [21]. After transcribing the written questionnaire and interview content, they were read repeatedly and open-coded in search of meaningful content. After coding and re-reading the data, grouping and categorizing similar concepts, and abstracting semantic units for each subject in consideration of their commonality and relevance, derivation of the core subject was performed. Data analysis was conducted simultaneously with data collection, and four researchers discussed whether the analyzed concept or category was consistent with the participants’ statements. Following this, the results of the qualitative analysis were verified by a professor who majored in qualitative research.
- Discussion
The first paragraph on page 9, "Through the ..." needs further support from the literature. Besides, aspects related to the evaluation of training should be studied in-depth, especially, some strategies aimed at improving the training that has been carried out, and possible problems in applying training to the job could be pointed out. Study limitations should be indicated in a separate section. Besides, literature should be provided that would open new lines of research.
Response: (L405-426) I have revised the content taking your comment into account.
- (Before) The limitations of this study are that the educational program is only a one-time and short-term effect, and the training time, the number of instructors, and the simulation situations are limited. This study proposes the following directions for future studies. Despite the confirmation of the NPPV simulation program being an effective intervention for increasing nursing knowledge and self-efficacy, further studies on long-term effects are needed. Second, based on the results of this study, developing various simulation scenarios related to ventilator nursing and to evaluate their effectiveness is necessary. In particular, in view of the characteristics of adult education, we propose that a high-quality educational program development and effectiveness evaluation study be conducted multiple times rather with sufficient education time and instructors who are familiar with simulation education
- (After) The limitations of this study are as follows. First, in terms of the composition of the educational program, the limitations identified through the written evaluation were short training time, limited simulation situation, and fewer instructors. In terms of the operation of the program, there were limitations in that it was a one-time short training and not a regular training program, and that the training hours were limited to the hours immediately after work. In terms of research, the limitations were that short-term knowledge and self-efficacy were measured without examining nurses’ practical performance to confirm the effectiveness of the program, and the dropout rate of the participants was high due to the limitation of nurses’ working hours. Through this, the following follow-up research is suggested. First, it was confirmed that the NPPV simulation program was an effective intervention in increasing nursing knowledge and self-efficacy; however, a future study on repetitive education and its long-term effects is needed because the training effect is reduced after a period. Second, based on the results of this study, we propose the development of various simulation scenarios and effectiveness evaluation studies related to ventilator nursing. In particular, given the nature of adult education, we propose implementing several high-quality educational programs and conducting effectiveness assessment studies through instructors familiar with sufficient training time and simulation training. Third, the measurement tools in this study were created by extracting items suitable for NPPV, which are primarily applied by nurses in general wards; hence, there is a limit to applying these tools to overall ventilator care, including the care of those subjected to invasive ventilation. Therefore, in future, we propose studies on simulation based learning including various case-based emergency and care for invasive ventilators.

Reviewer 3 Report
This paper aims to analyze an experiment (using control and experiment group) of the effectiveness of non-invasive positive pressure ventilation (NIIPV) ventilation simulation program for general ward. The analyses can be relevant, although I believe more analyses should be done, and the paper should be better framed and written. I provide more details about it:
- The sample is very low. How would you ensure that these results could be generalizable to a larger sample?
- Writing is improvable. There are many spaces between spaces missing, and English is also improvable. Authors should carefully proof-read the document.
- Section 1 is very short and there are not enough references of other works to justify the contribution. If there was a related work section, it could be fine, but as there is not, the background is not enough, and more references are needed. It would be very important to review what has been done in the literature to justify your contribution.
- Your contribution is not currently justified. You mention what you want to do, but there is not enough evidence based on previous work,
- There are very few current references and many of them are old. In fact, there is not any single reference from 2019 and 2020. This makes it difficult to justify the novelty.
- You mention that there were 20 sheets of written evaluation papers after the simulation program, but there are not many details about the questions there. You should provide more details.
- You should specify the year (not only months and dates) when the experiments were carried out.
- L183-L189: You should improve your writing as it is difficult to understand by reading the text.
- Table 2 and 3. You include many variables, but these variables have not been presented in the methodology. I believe the methodology is not properly framed to what you present later in the paper.
- Qualitative data analysis is not very comprehensive. You should try to elaborate more on this and enhance the discussion based on the particular statements.
- Self-efficacy should be more related to self-regulated learning (SRL) as SRL is a key factor in learning and there may be other SRL skills that you could comment on based on the qualitative analyses.
- In general, I believe that more analyses should be done to make this paper more solid.
Author Response
Reply to Reviewers:
Thank you for the comments and helpful suggestions regarding our manuscript. We have modified the manuscript accordingly and the revised details are provided below.
Reviewer reports:
Reviewer 3
Comments and Suggestions for Authors
This paper aims to analyze an experiment (using control and experiment group) of the effectiveness of non-invasive positive pressure ventilation (NIIPV) ventilation simulation program for general ward. The analyses can be relevant, although I believe more analyses should be done, and the paper should be better framed and written. I provide more details about it:
▪ The sample is very low. How would you ensure that these results could be generalizable to a larger sample?
Response: (L89-94) I have revised the content.
- (Before) The sample size for the quantitative study was calculated at a significance level of p<0.05, effect size of f=0.5, power of 1-β=0.80 using the G*Power 3.1.9.2 program, resulting in 27 people, with a dropout rate of 20%. In total, 34 people were recruited. The recruited subjects were assigned to 17 experimental groups and 17 control groups. Of these, five subjects were eliminated due to failure to participate in the scheduled training, and the final analysis subjects were 29 (15 in the experimental group and 14 in the control group).
- (After) The sample size for the quantitative study was calculated at a significance level of p<0.05, with an effect size of f=0.5 and power of 1-β=0.80 using the G*Power 3.1.9.2 program, resulting in 27 participants in each group, with a dropout rate of 10%. A total of 60 nurses were recruited. However, 31 nurses withdrew participation because of inability to participate due to workload. Of the 29 nurses who were finally recruited, 15 were assigned to the experimental group and 14 to the control group.
▪ Writing is improvable. There are many spaces between spaces missing, and English is also improvable. Authors should carefully proof-read the document.
Response: The entire paper has been proofread.
▪ Section 1 is very short and there are not enough references of other works to justify the contribution. If there was a related work section, it could be fine, but as there is not, the background is not enough, and more references are needed. It would be very important to review what has been done in the literature to justify your contribution.
▪ Your contribution is not currently justified. You mention what you want to do, but there is not enough evidence based on previous work,
▪ There are very few current references and many of them are old. In fact, there is not any single reference from 2019 and 2020. This makes it difficult to justify the novelty.
Response: (L27-64) I have added the content and modified this text per you suggestion.
- (Before) Ventilators are traditionally applied with invasive positive pressure ventilation through artificial airways such as tracheostomy or endotracheal intubation. However, in the initial ventilator treatment, non-invasive positive pressure ventilation (NPPV) is increasingly applied as a method to reduce the risk of having to secure the airway through artificial airway intubation, as well as complications such as vocal cord injury, and infection due to invasive causes [1].
For successful ventilator treatment, the patient's underlying disease, physical condition, and emotional condition are important [2]. Patients who use ventilators often experiences feeling of loss of communication, loss of sleep, loss of self-regulation, loss of self-determination, loss of personality, the presence of pain and fear, and dehumanization [3-4]. In particular, in the case of patients who use home ventilators in general wards, most of them are conscious; therefore, providing emotional care in these wards is much more important than in intensive care units. However, in actual clinical settings, the patients’ care mainly focuses on physical care, often leading to the neglect of emotional nursing care [3].
Since general ward nurses have fewer educational opportunities and experience in the ability to use ventilators than nurses in intensive care units, skillful handling of ventilators is not an easy for the former. Various studies on the usefulness of NPPV have shown that the frequency of intubation in the trachea is low and there is a reduction in the length of hospitalization, readmission rate, and mortality rate [1,5-6]. However, for the application and maintenance of NPPV, the time and effort required by skilled medical staff are important [7-8].
In the UK and Australia, in order to maintain the continuity of nursing in intensive care units and general wards, respiratory specialist educate and support various types of treatment knowledge and skills for medical staff in general wards [9-10]. There have been studies on topics such as the experience of patients who applied the ventilator, the educational needs of patients, the actual status of ventilator training, the knowledge and education requirements, and the experience of patients with the ventilator for the intensive care staff. Few studies have evaluated the effects of NPPV application experience and simulation education for general ward nurses.
- (After) Ventilators are conventionally applied with invasive positive pressure through artificial airways such as in tracheostomy or endotracheal intubation in intensive care units [1-2]. However, due to the development of medical devices, and improvement in medical equipment quality and patient survival rates, the number of patients applying for home medical ventilators with excellent performance and portability is continuously increasing [2-5].
As the number of patients requiring ventilator treatment increased, hospitals faced a lack of beds in intensive care units and an increase in medical expenses in South Korea [2-5]. It means that under the judgement of medical staff, many patients who need ventilator treatment are receiving that in general wards after changing to NPPV [2-4].
As the use of home ventilators increases in general wards, the proportion and time of nursing work related to ventilators is also increasing [2-4]. However, general ward nurses have fewer educational opportunities and experience in the ability to use ventilators than nurses in intensive care units, thus making skillful handling of ventilators difficult for the former [1-4].
For the successful administration of ventilator treatment, not only are a patient's underlying disease and physical condition important but also their emotional condition [6-7]. Patients who use ventilators often experiences feeling of a loss of communication, sleep, self-regulation, self-determination, and personality, and the presence of anguish, fear, and dehumanization [7-9]. In particular, in the case of patients who use home ventilators in general wards, most of them are conscious; therefore, providing emotional care to patients in these wards is more important than providing that in intensive care units. However, in actual clinical settings, patients’ care mainly focuses on physical care, often leading to the neglect of emotional care [7-8].
Various studies on the usefulness of ventilator treatment and non-invasive positive pressure ventilation (NPPV) have shown that the frequency of intubation in the trachea is low, and there is a reduction in the length of hospitalization, readmission rate, and mortality rate [9-11]. However, for the application and maintenance of NPPV, the time and effort required by skilled medical staff are significant factors [7, 13-14].
In the UK and Australia, to maintain the continuity of nursing in intensive care units and general wards, respiratory specialists educate and provide knowledge on various types of treatment and skills necessary for medical staff in general wards [15-16]. In South Korea, mainly in large hospitals, there are nurses in charge who assist in the application and management of home ventilators in general wards, and provide education for the medical staff regarding ventilators. However, there are few studies on the methods and effects of such an educational program. Most studies conducted in Korea focus on the problems and educational needs of ward nurses resulting from a lack of ventilator education, and an education program for ward nurses is necessary [2-4].
▪ You mention that there were 20 sheets of written evaluation papers after the simulation program, but there are not many details about the questions there. You should provide more details.
Response: The 20 sheets of written evaluation papers received after the simulation program included a questionnaire representing educational satisfaction; Figure 1 was revised because nurses’ feelings after the education program were not included in the qualitative data.
▪ You should specify the year (not only months and dates) when the experiments were carried out.
Response: In line 161, we mentioned the dates (from September 21 to September 28, 2018).
▪ L183-L189: You should improve your writing as it is difficult to understand by reading the text.
Response: (L231-236) I have modified this text.
- (Before) The work experience of the subjects varied from less than 1 year to more than 10 years, with most of them having less than 3 years of experience (72.4%). In terms of previous ventilator training, majority of the participants (82.8%) had previously received ventilator training. Regarding the experience of respiratory nursing within one year, most of the participants had none to nine experiences (Table 1).
- (After) The clinical work experience of the subjects varied from less than one year to more than ten years, with most participants having less than three years of clinical work experience (72.4%). In terms of whether they had previous experience in ventilator training, a majority (82.8%) of the participants had received ventilator training in the past. Regarding the number of ventilator nursing experiences within one year, most of the participants had none or less than nine years of experience (Table 1).
▪ Table 2 and 3. You include many variables, but these variables have not been presented in the methodology. I believe the methodology is not properly framed to what you present later in the paper.
Response: (L112-124) I have added the content you suggested in the text.
- (Before) The quantitative research tool of this study was prepared by referring to the safe initiation and management of mechanical ventilation [11] among the ventilator nursing interventions of adult nursing [12] and the Clinical Practice Guidelines of the American Association for Respiratory Care (AARC). It was produced through the process of revision 98 with expert advice.
- (After) The quantitative research tools of this study were prepared by referring to safe initiation and management of mechanical ventilation in clinical practice guidelines [17] of the American Association of Respiratory Therapy (AARC) and the ventilator nursing interventions for adult nursing [18]. Among them, general ventilator nursing and NPPV management were extracted, except for invasive ventilator management. A respiratory education nurse with more than seven years of experience consulted with one respiratory physician and prepared a preliminary questionnaire. Thereafter, the questionnaire was modified and supplemented with consultations from three nurses with master’s degrees and advanced practice nurses with over five years of clinical experience in a general hospital. Finally, the questionnaire was verified through a group of experts, including two head nurses with several years of experience in ventilator nursing, one professor from the Department of Adult Nursing, and one professor from the department of respiratory medicine.
Response: (L126-129) I have added the content you suggested in the text.
- (Before) As for the nursing ventilator related knowledge measurement tool, two head nurses with significant experience in ventilating nursing got the questions filled out by the following individuals: a nurse in charge of breathing education, one professor in the Department of Adult Nursing, and one professor in the department of respiratory medicine to evaluate the validity of the questions and correct parts with ambiguous meaning.
- (After) Ventilator nursing knowledge consisted of 10 preliminary questions regarding knowledge necessary for practical performance, such as preparation, maintenance, and management of ventilators; the items that needed semantic separation were separated and described during content validity verification.
Response: (L135-138) I have added the content you suggested in the text.
- (Before) The self-efficacy measurement tool on a ventilator was corrected by a nurse in charge of breathing education. This was done after filling out the questionnaire by two general ward head nurses with significant experience in ventilating nursing, one adult nursing professor, and one respiratory internal medicine professor.
- (After) The self-efficacy in ventilator nursing was evaluated through 15 preliminary questions, including on confidence, self-regulation, and task difficulty preference at each stage of evaluation, diagnosis and planning, performance, and evaluation of the ventilator nursing process. The final 15 questions were selected through content validation.
▪ Qualitative data analysis is not very comprehensive. You should try to elaborate more on this and enhance the discussion based on the particular statements.
Response: (L272-356) I have added detailed information taking your comment into consideration.
- (Before) As a result of analyzing the data for the development and effects of the simulation program, a total of 51 major statements were extracted. Those with repeated or similar meanings were categorized into 20 subcategories. Finally, three key words that included categories were derived: “to understand the psychological and physical discomfort of the respirator patient,” “there is practical help in the intervention of artificial respirator nursing,” and “development of artificial respirator nursing education and establishment of a strategy” (Table 4).
- (After) As a result of analyzing the data for the development and effects of the simulation program, a total of 51 major statements were extracted. Those with repeated or similar meanings were categorized into 20 subcategories. Finally, three key phrases related to the categories were derived: “to understand the psychological and physical discomfort of the patient applying NPPV,” “there is practical help available in ventilator care” and “development of ventilator training education and establishment of a strategy.”
Regarding the first theme—understanding the psychological and physical discomfort experienced by ventilator-applied patients—the nurses expressed that they experienced and understood the psychological and physical discomfort through the simulation program. On the sub-themes of experiencing psychological rejection, subjects who experienced NPPV expressed feelings of embarrassment, fear, anxiety, and frustration: “I heard it was slight pressure, but the air came out hard in my face, and I was embarrassed,” “I felt frustrated and trapped, and I couldn’t control it, so I think I got a temper later,” and “I was anxious because I couldn’t speak when I was wearing a mask.” Regarding the nurses’ experience of physical discomfort, they complained of physical discomfort such as when wearing heavy masks, pain in the face, and feeling nauseated while undergoing NPPV. “My stomach was full and I felt like throwing up” and “The tightness of the face was more painful and made it difficult to breathe.” Furthermore, the sub-themes of understanding patients with ventilators were expressed when the nurses understood the patients through their experience of NPPV. “Every time I ask the patient to be patient… I will understand their hearts a little,” “Sometimes I was annoyed because I couldn’t understand that NPPV patients take off their masks because they feel frustrated and uncomfortable, but when I applied it to myself, I understood 99% of their mind.”
The second theme was that the simulation was helpful in the intervention for ventilator nursing in clinical practice. After participating in the simulation program, they expressed that their fear was reduced and confidence was gained when seeing patients with ventilators in clinical practice. “After the experience, I saw a patient applying a ventilator, but I felt more confident than before,” “It was nice to be able to do it myself, so I could reduce my fear of NPPV,” and “I don’t think I will be embarrassed even if a patient with NPPV comes.” The other sub-theme was realizing the nursing needs of patients while applying NPPV. Nurses expressed that they realized the need for an explanation and emotional support when applying NPPV through the simulation program. “Every time I tell a patient to be patient, I will understand their mind a little. I thought I had to explain it well in the future,” “It was stuffier than I thought. It seems that a detailed explanation is needed before applying it to the patient,” “There were many times when I changed the water from the humidifier without saying anything, and I said that I had to endure it whenever I was having a hard time, but I also thought and reflected that I should provide more emotional support.”
The last theme is to establish a future ventilator training strategy. Nurses freely expressed the merits and improvement directions of this simulation program. Regarding the advantages of such a simulation program, the nurses expressed that it was practically helpful to perform the necessary manipulations directly through the machine used in the ward and receive feedback in situations that may occur in actual clinical settings; they further stated that it was interesting to have an opportunity to understand patients through their experience. “It was easy to understand by using the machine used in the actual ward. I was familiar with it a lot, but it was nice to learn about the part I did not exactly know,” “I liked the fact that I was able to better understand the patients while experiencing it in person, and it was nice to give an appropriate example,” “It was nice to be able to experience things that are not familiar to me. It’s good to have hands-on time to operate the machine yourself.” The second sub-theme is about the notable challenges faced in the program. The nurses expressed regret that the simulation program involved a limited situation in a short time. They also expressed that if there are additional practical instructors, more efficient education will be possible. “Various patient cases... How did they apply and what happened in a certain situation? What happened to the ABGA value and how did it change? If there was an education about...,” “The time was too short, so it seemed a tight schedule,” and “I wish there were more practicing instructors” (Table 4).
▪ Self-efficacy should be more related to self-regulated learning (SRL) as SRL is a key factor in learning and there may be other SRL skills that you could comment on based on the qualitative analyses.
Response: (L405-426) To determine the actual effect, the degree of performance directly applied to the patient should be checked, but the fact that it is not possible is a limitation of this study. I have added the limitations of this study under Conclusions.
- (Before) The limitations of this study are that the educational program is only a one-time and short-term effect, and the training time, the number of instructors, and the simulation situations are limited. This study proposes the following directions for future studies. Despite the confirmation of the NPPV simulation program being an effective intervention for increasing nursing knowledge and self-efficacy, further studies on long-term effects are needed. Second, based on the results of this study, developing various simulation scenarios related to ventilator nursing and to evaluate their effectiveness is necessary. In particular, in view of the characteristics of adult education, we propose that a high-quality educational program development and effectiveness evaluation study be conducted multiple times rather with sufficient education time and instructors who are familiar with simulation education
- (After) The limitations of this study are as follows. First, in terms of the composition of the educational program, the limitations identified through the written evaluation were short training time, limited simulation situation, and fewer instructors. In terms of the operation of the program, there were limitations in that it was a one-time short training and not a regular training program, and that the training hours were limited to the hours immediately after work. In terms of research, the limitations were that short-term knowledge and self-efficacy were measured without examining nurses’ practical performance to confirm the effectiveness of the program, and the dropout rate of the participants was high due to the limitation of nurses’ working hours. Through this, the following follow-up research is suggested. First, it was confirmed that the NPPV simulation program was an effective intervention in increasing nursing knowledge and self-efficacy; however, a future study on repetitive education and its long-term effects is needed because the training effect is reduced after a period. Second, based on the results of this study, we propose the development of various simulation scenarios and effectiveness evaluation studies related to ventilator nursing. In particular, given the nature of adult education, we propose implementing several high-quality educational programs and conducting effectiveness assessment studies through instructors familiar with sufficient training time and simulation training. Third, the measurement tools in this study were created by extracting items suitable for NPPV, which are primarily applied by nurses in general wards; hence, there is a limit to applying these tools to overall ventilator care, including the care of those subjected to invasive ventilation. Therefore, in future, we propose studies on simulation based learning including various case-based emergency and care for invasive ventilators.
Response: (L2-4) I have revised the title taking your comment into account.
- (Before) Title: Developing and Analyzing the effectiveness of a Non-Invasive Positive Pressure Ventilation (NPPV) Simulation Program for General ward Nurses
- (After) Title: The Effect of a Non-Invasive Positive Pressure Ventilation Simulation Program on General Ward Nurses’ Knowledge and Self-efficacy
▪ In general, I believe that more analyses should be done to make this paper more solid.
Response: As suggested by the reviewers, more analysis data were added, and the paper was further strengthened by referring to the latest literature.

Round 2
Reviewer 1 Report
This reversion paper has shown a great improvement regarding to the content coherence and the quality of research methodology , findings.
only very small parts can be corrected: for example line 147-157 suggest should be re-wording, line 278-326 also need reduce the length of sentence,to make it concise and clear.
Author Response
Reply to Reviewers: Thank you for the comments and helpful suggestions regarding our manuscript. We have modified the manuscript accordingly and the revised details are provided below.
Reviewer reports:
Reviewer 1This reversion paper has shown a great improvement regarding to the content coherence and the quality of research methodology, findings.Only very small parts can be corrected: for example line 147-157 suggest should be re-wording, line 278-326 also need reduce the length of sentence to make it concise and clear.
Response: (L153-163)I have modified the text, as you suggested.(Before) Based on previous studies [19-20], the experience of patients with NPPV was included inthe simulation program as a way of improving communication in the process of understanding the fear and rejection experienced by patients, and the formation of empathy. The draft of the program was based on prior studies on the educational needs of home ventilator nursing for general ward nurses and on errors and emergencies related to ventilator nursing [2-4]. A total of 32 educational needs were extracted in the survey contents [15]; respirator nursing practices frequently performed in general wards and errors with a tendency to occur frequently were listed according to the experience of the respiratory education nurse. Finally, one respiratory education nurse and three nurses in charge of education selected items suitable for general ward situations, andthree head nurses in a general ward and one professor from the department of respiratory medicine reviewed and selected six key details, including connecting and management of circuits, explaining the need for a ventilator and use of masks, ventilator manipulation, and treatment of clinical indicators.
(After) Previous studies on the educational needs of home ventilator nursing[3, 15] for general ward nurses and on errors and emergencies associated to ventilator nursing[2-4]were used as a basis to draft the program. Frequent ventilatornursing practices performed in general wards and recurring errors were included among the 32 educational needs that were drawn out from the survey contents based on the experiences of a respiratory education nurse. Appropriate items for the general ward situations were chosen by a respiratory nurse and three nurses who were in charge of education. Lastly, six details which includes connecting and management of circuits, explaining the need for a ventilator and use of masks, ventilator manipulation, and treatment of clinical indicators were reviewed and picked out by three head nurses in a general ward and a professor from the department of respiratory medicine.
Response:(L283-319) I have revised the text to take your comment into account.
(Before) Regarding the first theme—understanding the psychological and physical discomfort experienced by ventilator-applied patients—the nurses expressed that they experienced and understood the psychological and physical discomfort through the simulationprogram. On the sub-themes of experiencing psychological rejection, subjects who experienced NPPV expressed feelings of embarrassment, fear, anxiety ,and frustration: “I heard it was slight pressure, but the air came out hard in my face, and I was embarrassed,” “I felt frustrated and trapped, and I couldn’t control it, so I think I got a temper later,” and “I was anxious because I couldn’t speak when I was wearing a mask.” Regarding the nurses’ experience of physical discomfort, they complained of physicaldiscomfort such as when wearing heavy masks, pain in the face, and feeling nauseated while undergoing NPPV. “My stomach was full and I felt like throwing up” and “The tightness of the face was more painful and made it difficult to breathe.” Furthermore, the sub-themes of understanding patients with ventilators were expressed when the nurses understood the patients through their experience of NPPV. “Every time I ask the patient to be patient... I will understand their hearts a little,” “Sometimes I was annoyed because I couldn’t understand that NPPV patients take off their masks because they feel frustrated and uncomfortable, but when I applied it to myself, I understood 99% of their mind.”The second theme was that the simulation was helpful in the intervention for ventilator nursing in clinical practice. After participating in the simulation program, they expressed that their fear was reduced and confidence was gained when seeing patients with ventilators in clinical practice. “After the experience, I saw a patient applying a ventilator, but I felt more confident than before,” “It was nice to be able to do it myself, so I could reduce my fear of NPPV,” and “I don’t think I will be embarrassed even if a patient with NPPV comes.” The other sub-theme was realizing the nursing needs of patients while applying NPPV. Nurses expressed that they realized the need for an explanation and emotional support when applying NPPV through the simulation program. “Every time I tell a patient to be patient, I will understand their mind a little. I thought I had to explain it well in the future,” “It was stuffier than I thought. It seems that a detailed explanation is needed before applying it to the patient,” “There were many times when I changed the water from the humidifier with-out saying anything, and I said that I had to endure it whenever I was having a hard time, but I also thought and reflected that I should provide more emotional support.”The last theme is to establish a future ventilator training strategy. Nurses freely expressed the merits and improvement directions of this simulation program. Regarding the ad-vantages of such a simulation program, the nurses expressed that it was practically helpful to perform the necessary manipulations directly through the machine used in the ward and receive feedback in situations that may occur in actual clinical settings; they further stated that it was interesting to have an opportunity to understand patients through their experience. “It was easy to understand by using the machine used in the actual ward. I was familiar with it a lot, but it was nice to learn about the part I did not exactly know,” “I liked the fact that I was able to better understand the patients while experiencing it in person, and it was nice to give an appropriate example,” “It was nice to be able to experience things that are not familiar to me. It’s good to have hands-on time to operate the machine yourself.” The second sub-theme is about the notable challenges faced in the program. The nurses expressed regret that the simulation program involved a limited situation in a short time. They also expressed that if there are additional practical instructors, more efficient educationwill be possible. “Various patient cases... How did they apply and what happened in a certain situation? What happened to the ABGA value and how did it change? If there was an education about...,” “The time was too short, so it seemed a tight schedule,” and “I wish there were more practicing instructors” (Table 4).
(After) Regarding the first theme—understanding the psychological and physical discomfort experienced by ventilator applied patients—the nurses expressed that they experienced and understood thepsychological and physical discomfort through the simulation program. On the sub-themes of experiencing psychological rejection, subjects expressed feelings of embarrassment, fear, anxiety, and frustration: “I felt frustrated and trapped, and I couldn’t control it, so I think I got a temper later.” Regarding the nurses’ experience of physical discomfort, they complained of physical discomfort such as pain in the face and feeling nausea: “My stomach was full and I felt like throwing up” and “The tightness of the face was more painful and made it difficult to breathe.” Furthermore, the sub-themes of understanding patients with ventilators were expressed as follows: “Sometimes I was annoyed because I couldn’t understand that NPPV patients take off their masks because they feel frustrated and uncomfortable, but when I applied it to myself, I understood 99% of their mind.” The second theme was that the simulation was helpful in the intervention for ventilator nursing in clinical practice. After participating in the simulation program, they expressed that their fear was reduced and confidence was gained when caring patients with ventilators in clinical practice. “After the experience, I saw a patient applying a ventilator, but I felt more confident than before,” and “I don’t think I will be embarrassed even if a patient with NPPV comes.” The other sub-theme was realizing the nursing needs of patients while applying NPPV. Nurses expressed that they realized the need for an explanation and emotional support when applying NPPV through the simulation program: “Every time I tell a patient to be patient, I will understand their mind a little. I thought I had to explain it well in the future. The last theme is to establish a future ventilator training strategy. Nurses freely expressed the merits and improvement directions of this simulation program. Regarding the advantages of such a simulation program, the nurses expressed that it was practically helpful to perform the necessary manipulations directly through the machine used in the ward and receive feedback in situations that may occur in actual clinical settings; they further stated that it was interesting to have an opportunity to understand patients through their experience. “It was easy to understand by using the machine used in the actual ward. I was familiar with it a lot, but it was nice to learn about the part I did not exactly know.” The second sub-theme is about the notable challenges faced in the program. The nurses expressed regret that the simulation program involved a limited situation in a short time. They also expressed that if there are additional practical instructors, more efficient education will be possible: “Various patient cases... How did they apply and what happened in a certain situation? What happened to the ABGA value and how did it change? If there was an education about...” (Table 4)

Reviewer 3 Report
Authors have made a significant effort to improve the paper and they have included more qualitative analyses. These new analyses are relevant. However, I still have some comments to further improve the paper (although they are easier to address):
- Your paper is better framed but you may still need to add more references or add more details about the references you provide. For example, you indicate range [9-11] but you do not indicate specific details of each contribution. You may provide more details to better justify your contribution in relation to previous works.
- You say "Ventilator nursing knowledge consisted of 10 preliminary questions", but you then mention about 11 questions and you report 11 questions in the table. Could you clarify this mismatch or fix it?
- Your revisions on the qualitative analysis are very good and relevant. As the section is now longer, you may add a summarizing paragraph at the end of the section
- You acknowledge that it is difficult to obtain more results on SRL, but you may explicitly comment on this in the limitations.
Author Response
Reply to Reviewers:
Thank you for the comments and helpful suggestions regarding our manuscript. We have modified the manuscript accordingly and the revised details are provided below.
Reviewer reports:
Reviewer 31.Your paper is better framed but you may still need to add more references or add more details about the references you provide. For example, you indicate range [9-11] but you do not indicate specific details of each contribution. You may provide more details to better justify your contribution in relation to previous works.Response:(L61-73) Thank you for this insightful suggestion.I have added the content and modified the text you suggested.
(Before) However, there are few studies on the methods and effects of such an educational program. Most studies conducted in Korea focus on the problems and educational needs of ward nurses resulting from a lack of ventilator education, and an education program for ward nurses is necessary [2-4].
However, one existing study analyzed the causes of injuries related to home ventilators among patients in general wards of tertiary hospitals, and suggested that systematic and appropriate home ventilator management education for medical staff is necessary [2]. Two other existing studies found that general ward nurses had low nursing knowledge and emergency coping skills for patients with home ventilators, and that there was a high demand for education [3-4]. One of these studies suggests developing a systematic ventilator education program, while the other suggests practical education and monitoring systems based on clinical cases, or alternatively, using simulations [4]. Another study found that more than 70% of medical staff taking care of home ventilators in general wards did not receive systematic home ventilator–related education. These findings clearly illustrate the necessity of providing education to general ward medical staff and suggest that education—including problem-solving and effective analysis for the education program—should be conducted [15].
2.Yousay "Ventilator nursing knowledge consisted of 10 preliminary questions", but you then mention about 11 questions and you report 11 questions in the table. Could you clarify this mismatch or fix it? You acknowledge that it is difficult to obtain more results on SRL, but you may explicitly comment on this in the limitations.
Response: (L137-138) I have added the content and modified this text, as you suggested.
(Before) Ventilator nursing knowledge consisted of 10 preliminary questions regarding knowledge necessary for practical performance, such as preparation, maintenance, and management of ventilators; the items that needed semantic separation were separated and described during content validity verification. This tool consisted of 11 questions and evaluated on a 10-point scale,
(After) Ventilator nursing knowledge consisted of 10 preliminary questions regarding knowledge necessary for practical performance,such as preparation, maintenance, and management of ventilators; however, items requiring one semantic separation were split into two items during the content validation. Finally, this tool consisted of a total of 11 questions and evaluated on a 10-point scale,
3.Your revisions on the qualitative analysis are very good and relevant. As the section is now longer, you may add a summarizing paragraph at the end of the section
Response: (L320-323) I have added the contentto address your comment.
In summary, the simulation program experience was based on three themes: understanding the psychological and physical discomfort of patients receiving NPPV, practical help in the intervention of mechanical ventilator nursing, and establishing a strategy for developing education for mechanical ventilator nursing.
4.You acknowledge that it is difficult to obtain moreresults on SRL, but you may explicitly comment on this in the limitations.Response:(L412-414) Ihave added the contentto address your comment, thank you.
(Before) First, it was confirmed that the NPPV simulation program was an effective intervention inincreasing nursing knowledge and self-efficacy; however, a future study on repetitive education and its long-term effects is needed because the training effect is reduced after a period.
(After) First, it was confirmed that the NPPV simulation program was an effective intervention in increasing nursing knowledge and self-efficacy; however, self-efficacy is closely related to self-directed learning and further research is needed to clarify the relationship. In addition, a future study on repetitive education and its long-term effects is needed because the training effect is reduced after a period.
